# Physicochemical Characterization of Dextran HE29 Produced by the *Leuconostoc citreum* HE29 Isolated from Traditional Fermented Pickle

**DOI:** 10.3390/molecules28207149

**Published:** 2023-10-18

**Authors:** Hümeyra İspirli

**Affiliations:** Food Engineering Department, Engineering Faculty, Bayburt University, Bayburt 69000, Türkiye; humeyraispirli@bayburt.edu.tr; Tel.: +90-(0)-458-211-11-52

**Keywords:** traditional pickles, lactic acid bacteria (LAB), exopolysaccharides (EPS), dextran

## Abstract

In this study, lactic acid bacteria (LAB) strains were isolated from traditional fermented pickles, and among the identified strains, *Leuconostoc citreum* HE29 with a strong slimy colony profile was further selected to determine the physicochemical and techno-functional properties of its exopolysaccharide (EPS). Glucose was the only sugar monomer in the core unit of EPS HE29 detected by HPLC analysis, and glucan HE29 revealed 7.3 kDa of molecular weight. Structural characterization of glucan HE29 by ^1^H and ^13^C NMR spectroscopy analysis demonstrated that EPS HE29 was a dextran-type EPS containing 5.3% levels of (1 → 3)-linked α-D-glucose units. This structural configuration was also supported by FT-IR analysis, which also demonstrated the functional groups within the dextran HE29 structure. In terms of thermal properties detected by TGA and DSC analysis, dextran HE29 demonstrated a degradation temperature of around 280 °C, showing its strong thermal features. A semi-crystalline nature was observed for dextran HE29 detected by XRD analysis. Finally, AFM and SEM analysis revealed tangled network-like properties and web-like branched structures for dextran HE29, respectively. These findings suggest the importance of plant-based fermented products as LAB sources in obtaining novel EPS structures with potential techno-functional roles.

## 1. Introduction

Lactic acid bacteria (LAB) are the main microbial group conducting food fermentations, and, together with the role of LAB strains as starter cultures, the formation of distinct metabolites during the fermentation process makes this microbial group indispensable for food production. LAB strains can produce different metabolites that can improve the technological and functional features of food products, and exopolysaccharides (EPSs) are one of the main metabolites that can be involved in the improvement of both the technological and functional properties of food products. LAB strains can produce homopolymeric and heteropolymeric EPSs in terms of a structural configuration comprising only one type of sugar monomer and two or more types of sugar monomers in the EPS repeating unit, respectively. Both EPS types can be important for functional and technological purposes, although in terms of yield, production costs and practicability, homopolymeric EPSs can be advantageous in comparison to heteropolymeric EPSs as sucrose is the substrate required for homopolymeric EPS production. The main structure of homopolymeric LAB EPSs is the glucan-type EPSs containing only glucose in the core repeating unit, and glucose units can be linked as (1 → 6), (1 → 2), (1 → 3) or (1 → 4)-linked α-D-glucose units. The type and proportion of these linkages in the glucan structure determine the glucan type, and so far, four glucans have been characterized as dextran, alternan, reuteran and mutan formed by mainly (1 → 6)-linked α-D-glucose units, alternating between (1 → 6)/(1 → 3)-linked α-D-glucose units, mainly (1 → 4)-linked α-D-glucose units and mainly (1 → 3)-linked α-D-glucose units [1,2,3,4]. Apart from glucans, LAB strains were reported to produce fructan-type EPSs containing only fructose units in the EPS repeating unit structure [5]. Finally, the homopolymeric EPS type is the galactan-type EPS, and only a few studies have reported galactan production in LABs [6,7].

Recent studies revealed the importance of the source of the LAB strains to unveil novel EPS structures that can have important technological and functional roles for food fermentation systems as well as food technology [8,9]. In terms of the fermentation environment, pickled vegetables might be important sources for homopolymeric EPS producer LAB strains as sucrose presents within this fermentation environment. This fact was previously assigned to the sourdough environment as most of the sourdough isolate LAB strains were proposed to be homopolymeric glucan producers [10]. Importantly, the isolation of LAB strains from fermented plant-based food products can be not only interesting in terms of elucidating their microflora but also crucial for obtaining novel metabolites such as EPS. From this perspective, the aim of this study was to isolate and identify an EPS producer LAB strain from pickled vegetables and characterize the structural and technological functions of this EPS. For this, among the isolates, *Leuconostoc citreum* HE29 was further selected for its EPS production capabilities due to its strong slimy morphology. The sugar monomer of EPS HE29 was determined through HPLC analysis, whereas ^1^H and ^13^C NMR spectroscopy analysis was performed to characterize the core unit structure of EPS HE29. For the determination of the functional groups within the EPS HE29 structure, FT-IR analysis was applied. In terms of the thermal characterization of EPS HE29, TGA and DSC analysis were performed. To determine the physical status of EPS HE29, XRD analysis was applied, whereas for the characterization of the structural morphology as well as the structural topography of EPS HE29, SEM and AFM analysis were performed, respectively. This study could be important to unveil the importance of the LAB strains originating from plant-based fermented food products for the production of homopolymeric α-glucans.

## 2. Results and Discussion

### 2.1. Screening EPS Producer Strains, Their Isolation and Genotypic Identification

Home-made traditional pickled vegetables were used as a LAB source to detect the EPS production capabilities of these LAB strains. Purified colonies were spread to modified BHI agar plates containing sucrose as a C source, and four isolates appeared to form strong slimy colonies (Figure 1). Genotypic discrimination was applied to EPS producer isolates through RAPD-PCR analysis, and four isolates as distinct strains were further identified through 16S rRNA gene analysis. Strains VB113, VK1 and VK36 were identified as *Leuconostoc mesenteroides* strains, whereas strain HE29 was a *Leuconostoc citreum* strain. The latter strain was further selected to determine its EPS characteristics as a home-made pickle isolate *Leuconostoc citreum* strain. Previous studies also reported the potential of LAB strains from pickles with different origins for EPS production capabilities [11,12,13], and testing different traditional sources might result in the identification of distinct EPS producer LAB strains. From this perspective, *Leuconostoc citreum* HE29 was grown in a modified BHI medium containing 10% sucrose, and the extracted EPS was further characterized. The GenBank accession numbers of the 16S rRNA gene sequences of strains VB113, VK1, HE29 and VK36 were OQ801446-OQ801449.

### 2.2. Monomer Profile and Molecular Weight of EPS HE29

*Leuconostoc* spp. are one of the main homopolymeric glucan-type EPS producer strains among LAB strains with *Leuc. mesenteroides* being the well-known microbial species in this group, and recently, *Leuc. citreum* strains have attracted attention because they can be present in distinct sources as EPS producer strains. Similar to *Leuc. mesenteroides*, an important number of studies reported that glucan-type EPS, containing glucose as the only sugar monomer in the EPS repeating unit, was the main EPS type for distinct *Leuc. citreum* strains [14,15,16,17]. Additionally, levan production, as fructose with the only sugar in the EPS repeating unit structure, was demonstrated for the *Leuc. citreum*-BMS strain [18]. Considering the slimy profile of strain HE29 in a sucrose-rich environment suggesting its EPS production ability, at first, the sugar monomer profile of its EPS was determined through HPLC analysis. As can be seen in Figure 2A, the monomeric composition of EPS HE29 was only formed by glucose units, suggesting that *Leuc. citreum* HE29 was a glucan-type EPS producer.

The glucan yield of strain HE29 was found to be 7.04 ± 0.3 g L^−1^ when the modified BHI medium was used as the growth medium. This level of glucan production demonstrates the potential of *Leuc. citreum* HE29 for different strategies, whether for producing the glucan for ex situ applications or for in situ glucan production during food fermentations [19]. Previous studies revealed higher glucan production for *Leuc. citreum* strains up to 35 g L^−1^, and optimization of the glucan yield for strain HE29 will be tested as a future goal as glucan yield can be altered depending on culture and medium conditions [11,20].

In terms of the molecular weight of glucan HE29, a molecular weight of 7.3 kDa was calculated through GPC analysis (Figure 2B), and this level of molecular weight was in general lower in comparison to the other glucan molecular weights [21,22,23,24], but dextrans with similar molecular weights were also reported previously [25,26]. The relatively low level of molecular weight for glucan HE29 can be interesting for the applications of the development of non-viscous food formulations, especially for beverage formulations.

### 2.3. Structural Characterization of Glucan HE29 through NMR Analysis

Following the detection of glucose as the only sugar monomer in the EPS structure, ^1^H and ^13^C NMR analysis was applied to unveil the structure of glucan HE29. Figure 3A demonstrates the ^1^H spectrum of the glucan HE29 showing the spectral resonances between 3.3 to 4.1 ppm and 4.6 to 5.25 ppm reflecting the ring proton and anomeric regions, respectively [27,28]. Importantly, in the anomeric region, spectral resonances were observed at 4.85 and 5.2 ppm, which were characteristic of (1 → 6)-linked α-D-glucose units and (1 → 3)-linked α-D-glucose units, respectively, showing that glucan HE29 was a dextran-type polysaccharide comprising approximately 5.3% levels of (1 → 3)-linked α-D-glucose units [1,2]. As can be seen in Figure 3B, ^13^C NMR spectra of glucan HE29 also confirmed the dextran structure, as anomeric signals, which were previously assigned to (1 → 3)-linked α-D-glucose containing a dextran structure, were observed at 97.7, 73.2, 71.2, 70.2, 69.6 and 65.6 ppm, which corresponded to the C-1, C-3, C-2, C-5, C-4 and C-6 of the glucose residues, respectively [1].

Overall, both ^1^H and ^13^C NMR data revealed that *Leuc. citreum* HE29 as a home-made traditional pickled vegetable isolate produced dextran containing low levels of (1 → 3)-linked α-D-glucose units. This finding was in accordance with the literature data as distinct *Leuc. citreum* strains were reported to produce dextran-type EPSs, although the core dextran structures differed depending on the strain-specific conditions. For instance, *Leuc. citreum* strains NM105 from sauerkraut, SK24.002 from traditional pickled vegetables and B-2 from fermented pineapple were shown to produce a highly α-(1 → 2) branched dextran [14], a highly α-(1 → 3) branched dextran [11] and dextran containing α-(1 → 3)/α-(1 → 2)-linked d-glucopyranose units [15], respectively, suggesting the strain-specific conditions for the type of formed dextran in *Leuc. citreum* strains. Additionally, the alterations in the dextran structure of *Leuc. citreum* strains could be observed to be within the same origin as reported for the sourdough isolates [1]. These findings suggested that strain-specific conditions, potentially originating from the variations in the structure of glucansucrases, could determine the final dextran structure produced by distinct LAB strains.

### 2.4. FT-IR Analysis of Dextran HE29

The FT-IR spectra of dextran HE29 were recorded within the 4000–400 cm^−1^ range to determine the presence of distinct functional groups in the dextran HE29 structure (Figure 4). A characteristic wide peak between 3000 and 3400 cm^−1^ demonstrating the hydroxyl groups within the polysaccharide structure was observed, suggesting that the tested material was a polysaccharide [29,30,31,32]. This wide peak was followed by two peaks observed at 2885 cm^−1^ and 1635 cm^−1^, which were indicative of the C-H bond stretching vibrations and existence of the carboxylic groups within the sugar ring, respectively [7,29]. A sharp peak at around 1005 cm^−1^ originating from the C-O-C and C-O atomic stretchings was observed, which was also characteristic for polysaccharides, and importantly, this peak demonstrated the existence of the (1→6)-linked α-D-glucose units within the structure of dextran HE29 [29]. In accordance with the previous data, three peaks were observed at around 1350, 850 and 550 cm^−1^ in the FT-IR spectra of dextran HE29, which were suggested to be related to the (1→3)-linked α-D-glucose units [11,33] in the dextran structure. Overall, FT-IR spectra of dextran HE29 demonstrated the presence of the functional groups within its structure, and, importantly, an integration between the FT-IR and NMR spectra was observed, confirming the dextran HE29 structure as a dextran containing (1 → 3)-linked α-D-glucose units.

### 2.5. Thermal Properties of Dextran HE29

One of the important processes in food preparation is the thermal application and utilization of components with strong thermal features in food formulations, which is critical in terms of technological and functional perspectives. Homopolymeric EPSs are good examples with strong thermal properties, and the structural characteristics of EPSs can be determinant of the final thermal features of EPSs [11,19,29,34]. In this regard, the thermal properties of dextran HE29 were recorded using TGA and DSC analysis. Figure 5A demonstrates the TGA analysis of dextran HE29 recorded dynamically as weight loss during the temperature increment. A prior weight loss up to 100 °C at around 8% was observed originating from the moisture loss of the dextran HE29 [35]. This phase as a mild degradation period was observed up to 260 °C, and the gradual decrement of the weight during the temperature increment in this period was suggested to be a positive characteristic of EPSs as they might hold increased levels of water in the food formulations [36]. This phase was followed by a sharp mass loss phase (around 30% mass loss was recorded) observed to be between 260 and 320 °C, revealing the thermal degradation of dextran HE29 [11,20]. Previously, similar degradation profiles were also recorded for other homopolymeric EPSs [20,28,29,36], and the reason for this sharp degradation profile was suggested to be related to the C-O and C-C bonding deformations in sugar rings [37]. Finally, a gradual mild weight loss was observed following the second degradation phase, and around 30% of the dextran HE29 remained as a residue at 500 °C. Consistent with the TGA analysis of dextran HE29, the DSC analysis revealed the smooth degradation profile of dextran HE29 tested between 0 and 500 °C, demonstrating the strong thermal profile of this homopolymeric EPS (Figure 5B) [23]. Overall, testing the thermal characteristics of dextran HE29 using TGA and DSC analysis showed that dextran HE29 had strong thermal features with a degradation temperature of around 280 °C, suggesting the potential technological and functional roles of this dextran for high-thermal food applications.

### 2.6. Crystallographic Properties of Dextran HE29

The crystallographic nature of EPSs is an important factor in understanding their physicochemical properties, and, in this regard, XRD analysis was applied to dextran HE29 to unveil whether this polymer has an amorphous or crystalline nature. As can be seen in Figure 6, characteristic mostly weak peaks, in 10 to 40 spectra of 2θ value, were observed in the XRD spectra of dextran HE29, and this pattern was previously associated with a semi-crystalline nature of EPSs [7,37]. Previous studies reported that the physical status of homopolymeric EPSs can be amorphous [23], crystalline [24] or semi-crystalline [7,37] like dextran HE29, suggesting the potential function of the EPS structures in their final crystallographic nature. Importantly, the semi-crystalline nature of polymers was previously suggested to be a positive factor in terms of acting as stabilizer/thickener compounds for food formulations [7], and dextran HE29 as a semi-crystalline EPS might find applications as a thickener agent.

### 2.7. Atomic Force Microscopy Analysis of Dextran HE29

The topographical morphology of dextran HE29 was tested by using AFM analysis, and Figure 7 demonstrates the morphological properties of the aqueous solution of dextran HE29 recorded following the air drying at RT. This image reveals the formation of the many small ellipsoidal or spheroidal particles at different heights, supporting the fact that dextran HE29 might form tangled network-like structural features in its aqueous solution as suggested previously [29,38,39]. Importantly, the formation of these tangled network-like properties for an aqueous solution of EPSs was suggested to be a positive factor in holding water, and, in terms of food formulations, this characteristic of dextran HE29 might result in it acting as a potential biothickener agent [29,39]. Overall, AFM images of dextran HE29 revealed the formation of small ellipsoidal/spheroidal lumps, which potentially suggest the strong capability of dextran HE29 to bind water in a food formulation.

### 2.8. Scanning Electron Microscopy Analysis of Dextran HE29

The microstructural properties of dextran HE29 were determined through SEM analysis, and, as can be seen in Figure 8, compact web-like branched structures were observed for dextran HE29. In previous studies, these types of web-like compact microstructural features of distinct EPSs were associated with the potential enhancement of the physicochemical properties of food products in terms of rheological features [29,40]. At relatively lower magnification levels, this compact morphology appeared to show fiber-like properties, which also potentially support the role of dextran HE29 in improving the rheological characteristics in food systems as previously noted [40]. Overall, SEM analysis of dextran HE29 clearly demonstrated its compact fiber-like microstructure, and studies are underway to determine the physicochemical role of dextran HE29 in water solutions as well as in food formulations as a potential biothickener agent.

## 3. Materials and Methods

### 3.1. Collection of Pickle Samples and Isolation of LABs

Traditional pickle samples were collected from four households in the Bayburt and Ankara provinces of Türkiye. The collected pickles were immature melon, common bean, red beet and pear pickled and produced under traditional conditions. These four pickle samples were then used as the materials for the isolation of LAB strains. For this, standard serial dilutions were applied to the De Man, Rogosa and Sharpe (MRS) agar, and plates were incubated at 37 °C for 2 days under anaerobic conditions. Following the incubation period, colonies were tested by using Gram staining and the catalase test, and potential LAB strains with distinct morphological features were subjected to molecular discrimination and identification tests.

### 3.2. Genotypic Characterization of Isolates Using Rep-PCR and Bacterial Identification of EPS Producers

In total, 60 isolates were selected with different morphological features, and the genomic DNA of the isolates was extracted using the phenol-isoamyl-alcohol methodology. When required, the Genomic DNA extraction kit (Invitrogen, Waltham, MA, USA) was also used following the manufacturer’s instructions. For the genotypic discrimination of the isolates, rep-PCR analysis using GTG5 primer (5′-GTGGTGGTGGTGGTG-3′) was applied using a previously described methodology [41]. Genotypically discriminated isolates were then tested for their EPS production capabilities depending on their slimy profiles in modified BHI agar plates as described earlier [28], and selected EPS producers were identified through 16S rRNA gene sequencing as described previously. Briefly, PCR reactions were set using universal primers AMP_F (5′-GAGAGTTTGATYCTGGCTCAG-3′) and AMP_R (5′-AAGGAGGTGATCCARCCGCA-3′) [42] in order to amplify 1.5 kb of 16S rRNA gene, and the PCR reaction mixture contained other required components described elsewhere [41]. Following the confirmation of the amplification of the 16S rRNA gene by agarose gel electrophoresis, PCR products were sent to Medsantek (İstanbul, Turkey) for sequencing, and isolates were identified through BLAST analysis with a similarity level of 99–100%.

### 3.3. EPS Extraction from Leuconostoc citreum HE29, Monomer Profile and Molecular Weight Determination

For the extraction of EPS, *Leuconostoc citreum* HE29 was grown in a modified BHI medium [29] containing up to 3% sucrose for 2 days at 37 °C without agitation. This modified BHI medium did not contain yeast extract as a source of β-glucan, which can interfere with the EPS during the extraction process. At first, *Leuconostoc citreum* HE29 was grown from its −80 °C glycerol stock in MRS broth, and from the overnight cultures, %1 was inoculated to a modified BHI medium. The grown cultures were then centrifuged, cells were removed, and EPS was then extracted from the culture supernatant by ethanol precipitation [34]. For this, a similar level of chilled ethanol was added to the culture supernatant of strain HE29, which was then subjected to overnight precipitation at 4 °C, and the precipitated EPS was collected by centrifugation. The crude EPS was dissolved in up to 100 mL of UP H_2_O with gentle heating, and the EPS solution was subjected to ethanol precipitation with 2× volume of chilled ethanol. EPS was recovered again as described above, and trichloroacetic acid (TCA) was added to the EPS solution at a 15% level for protein removal. Following the precipitation of proteins, the pH of the EPS solution was set to pH 7, and partially purified EPS was recovered by the addition of 3× volume of chilled ethanol to the EPS solution. A final centrifugation step was applied to obtain the partially purified EPS, which was then subjected to a dialysis step using a 12,000–14,000-Da membrane (Visking Dialysis Membrane, Medicell International, London, UK) for 72 h. A final lyophilization step was applied to the purified EPS, and the lyophilized EPS was stored at 4 °C for further analysis.

The monomer profile of EPS HE29 was determined by using HPLC (Shimadzu, Kyoto, Japan) analysis using a CHO column suitable for the detection of sugars, and rhamnose, glucose, galactose and fructose were the standard sugars used during the HPLC analysis. The details of the HPLC analysis were given elsewhere [24,29]. Briefly, EPS HE29 was hydrolyzed with 0.5 M H_2_SO_4_ at 95 °C for 12 h, which was followed by a neutralization step using 4 M NaOH. The hydrolysate was then given to the CARBOsep CHO-682 Pb column in an HPLC system (Shimadzu) with column conditions of 25 °C column temperature, a mobile phase of H_2_O and a flow rate of 0.7 mL min^−1^. The sugars were then detected with a RID-10A refractive index detector.

For the determination of the molecular weight of EPS HE29, gel permeation chromatography (GPC) analysis was applied at the TUBİTAK-MAM Research Centre (Gebze, Türkiye). For this, an Agilent 1260 Series instrument equipped with an RI detector was used, and the instrument conditions were a PL-aquagel-OH 8 μm Mixed-H column, a mobile phase of UP H_2_O with a 1 mL m^−1^ flow rate and 25 °C column temperature. The calibration curve was set with polyethylene oxide standards (calibration interval of Mp = 109–1522.000 g mol^−1^), and the molecular weight of EPS HE29 was calculated against this calibration curve.

### 3.4. Structural Determination of EPS HE29 through NMR Analysis

For the determination of the structural profile of EPS HE29, ^1^H and ^13^C NMR spectroscopy analysis was applied using an Agilent 400 MHz NMR instrument, and details of the NMR analysis were given elsewhere [29].

### 3.5. Determination of the Function Groups within EPS HE29 Structure by FT-IR Spectroscopy Analysis

A Perkin Elmer Spectrum 65 FT-IR spectrophotometer was used for the FT-IR analysis. FT-IR spectra of EPS HE29 were scanned in the area of the 4000–400 cm^−1^ region at transmission mode, and 4 cm^−1^ of spectral resolution was applied during the scanning process [29].

### 3.6. Thermal Characterization of EPS HE29 through TGA and DSC Analysis

An EXSTAR 7300 (SII Nanotechnology, Minato-ku, Tokyo) thermal analyzer was used for the determination of the TGA profile of EPS HE29. For this, EPS HE29 was subjected to a heating process in the range of 25–800 °C temperature under a nitrogen atmosphere with a heating ramp of 10 °C min^−1^, and the thermogravimetric loss in the EPS HE29 was recorded. DSC analysis of EPS HE29 was performed with a PerkinElmer DSC 8000 instrument using a previously described methodology [29].

### 3.7. Crystallographic Analysis of EPS HE29 through XRD Analysis

The physical and crystallographic properties of EPS HE29 were determined through XRD analysis using a Bruker D8 Discover instrument [29]. For this, Cu Kα radiation, using a Ni filter with generator settings of 40 kV and 40 mA, was performed, and XRD instrument settings and a Breg Brentano θ:2θ geometry application were applied to obtain the XRD diffractogram. The scan for the diffraction was applied within 5–90° in the 2θ range, and 0.03° increments in the step size were used.

### 3.8. Morphological Characterization of EPS HE29 through SEM and AFM Analysis

SEM analysis was applied to observe the microstructural features of EPS HE29 using a FEI Nova (Nano SEM 450) instrument. For this, a fixation step of EPS HE29 to the SEM stubs was applied followed by the coating step of stubs with a ~10 nm gold layer. During SEM analysis, 5.0 kV of accelerating voltage was applied, and images were taken under a 2000 to 50,000× magnification level.

The surface topographical properties of EPS HE29 were determined through AFM analysis using a previously described methodology [39]. Briefly, an AFM PLUS+ (NanoMagnetic) instrument was used for AFM analysis using a silicon (Si) tip (a radius of approximately 10 nm). A scan rate of 5 μm/s was applied with a resolution of 256 × 256 points, and images were obtained under first-order flattening.

## 4. Conclusions

Fermented vegetables can be important reservoirs for LAB strains with techno-functional roles. In this study, *Leuc. citreum* HE29 as an EPS producer strain was isolated from traditional pickled vegetables, and its EPS was characterized. EPS HE29 was a dextran-type homopolymeric EPS containing approximately 5.3% levels of (1 → 3)-linked α-D-glucose units. A molecular weight of 7.3 kDa was observed for dextran HE29. In terms of technological functions, dextran HE29 demonstrated strong thermal properties, suggesting its suitability for high-thermal applications of food processing. Additionally, dextran HE29 demonstrated a semi-crystalline nature, tangled network-like properties in its aqueous solution and a compact fiber-like microstructure detected using XRD, AFM and SEM analysis, suggesting its suitability to be used as a biothickener agent in food formulations. Future studies will focus on determining the physicochemical and bioactive functions of dextran HE29, especially with regard to its role in the production of fermented vegetables.

## Figures and Tables

**Figure 1 molecules-28-07149-f001:**
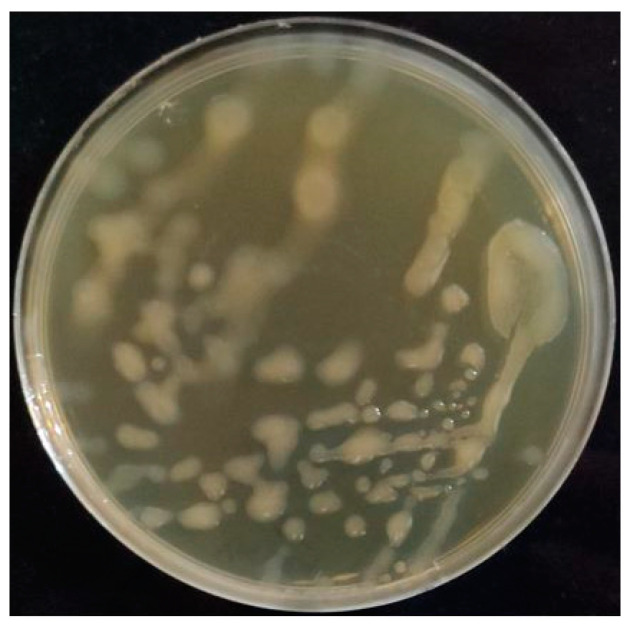
Slime colony morphology of *Leuc. citreum* HE29 strain on modified BHI agar medium containing sucrose.

**Figure 2 molecules-28-07149-f002:**
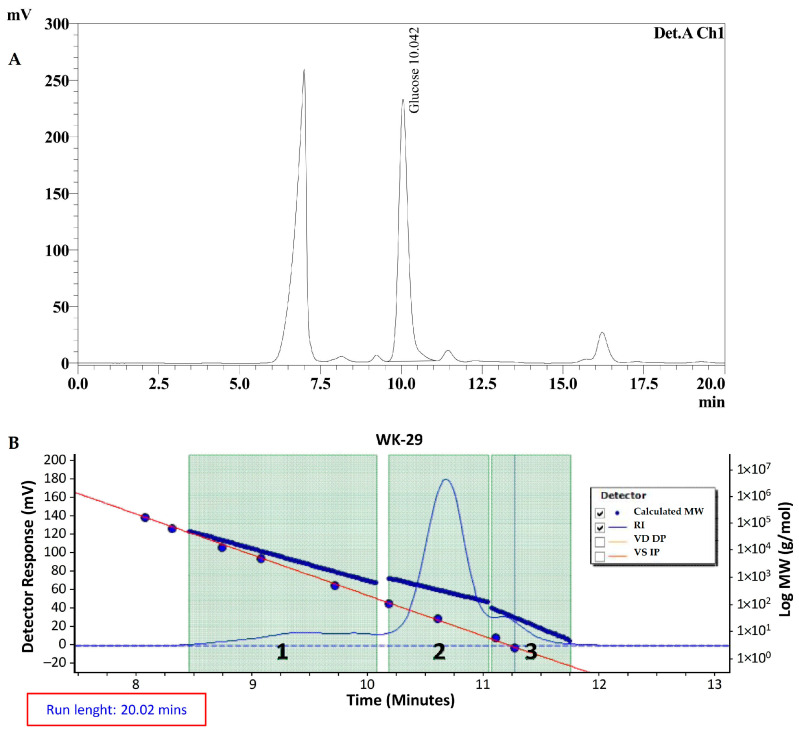
HPLC chromatogram of EPS HE29 demonstrating that glucose was the only sugar in EPS repeating unit of EPS HE29 (**A**). The first peak is the impurity salt formed during hydrolysis and neutralization step of EPS. GPC chromatogram of EPS HE29 showing the molecular weight of EPS HE29 as 7.3 kDa (**B**). Second peak originates from the EPS HE29, whereas the first and third peaks are the column impurity (**RI: Refractive Index, VD DP and VS IP: Viscometer signals**).

**Figure 3 molecules-28-07149-f003:**
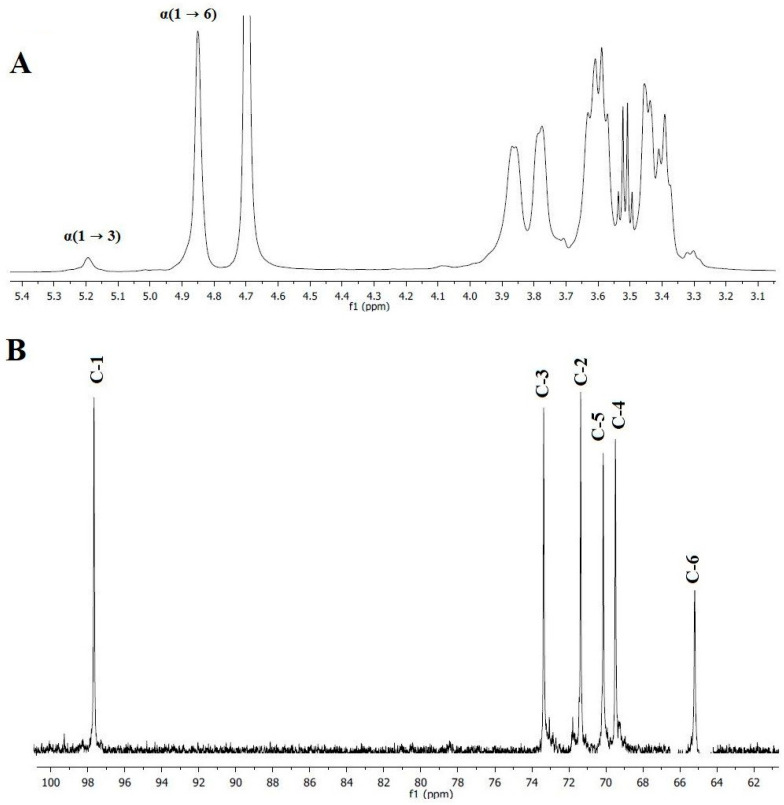
^1^H NMR (**A**) and ^13^C NMR (**B**) spectra of dextran HE29 produced by *Leuc. citreum* HE29.

**Figure 4 molecules-28-07149-f004:**
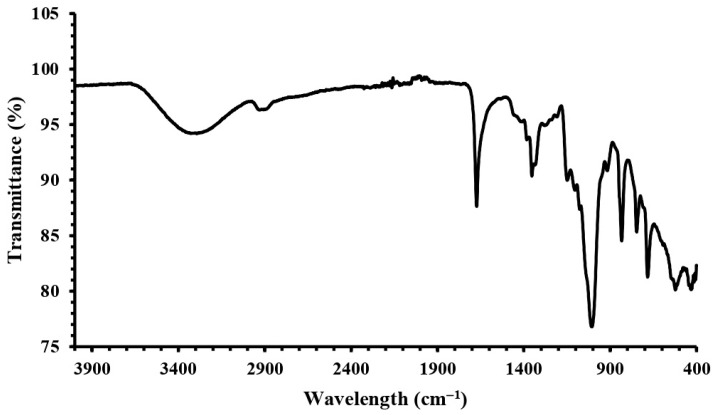
FT-IR spectra of dextran HE29 scanned in 400–4000 cm^−1^ range showing the presence of distinct functional groups within dextran HE29 structure.

**Figure 5 molecules-28-07149-f005:**
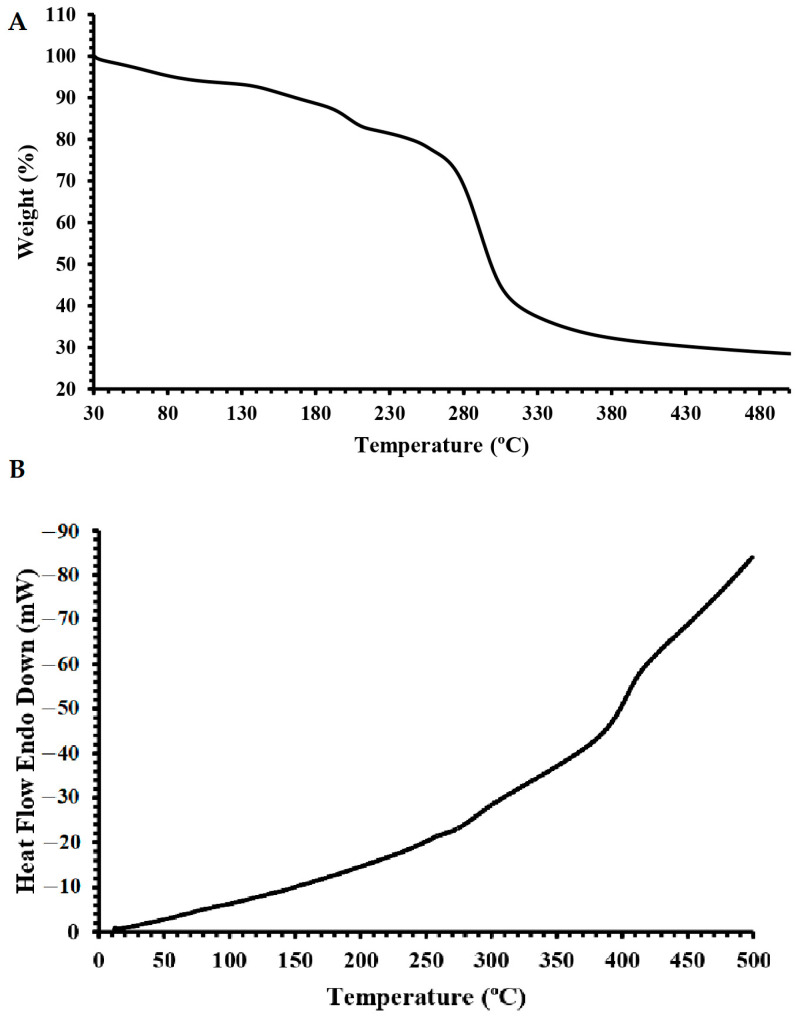
TGA (**A**) and DSC (**B**) thermogram of dextran HE29 revealing its high level of thermal stability.

**Figure 6 molecules-28-07149-f006:**
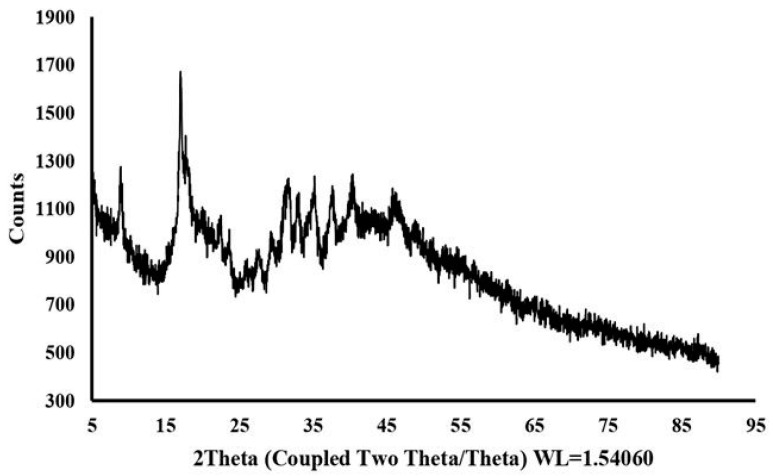
XRD spectra of dextran HE29 demonstrating semi-crystalline nature of dextran HE29.

**Figure 7 molecules-28-07149-f007:**
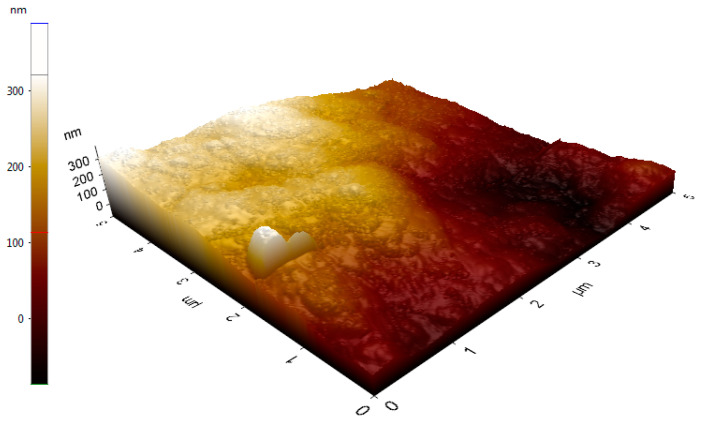
AFM image of dextran HE29 at cubic scale showing small ellipsoidal/spheroidal lumps within the dextran structure.

**Figure 8 molecules-28-07149-f008:**
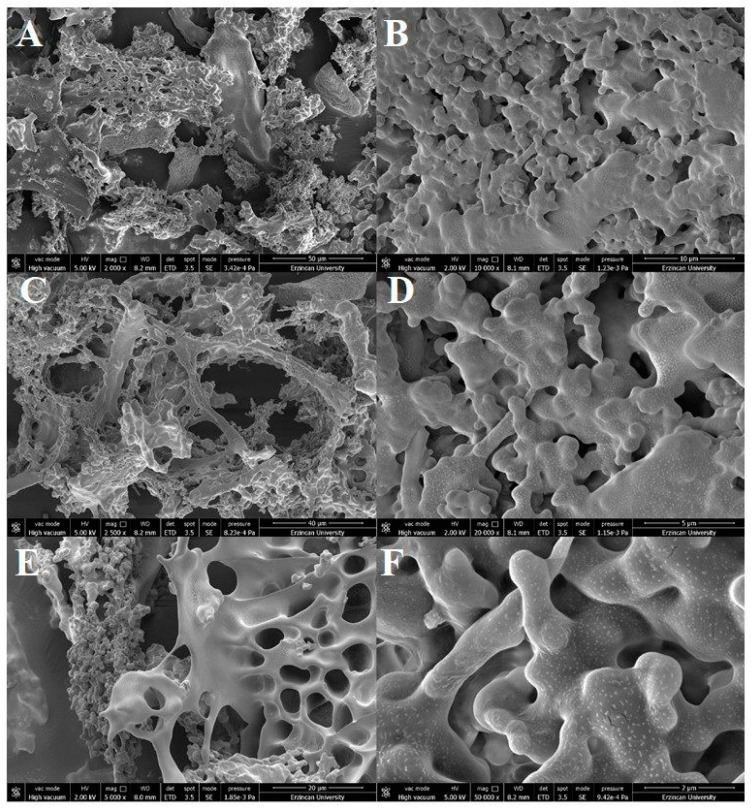
SEM images of the dextran HE29 at different magnification levels of 2000× (**A**), 10,000× (**B**), 2500× (**C**), 20,000× (**D**), 5000× (E) and 50,000× (**F**), demonstrating its compact web-like branched structural characteristics.

## Data Availability

All data are available in the article.

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
