# Peer review of "Physicochemical Characterization of Dextran HE29 Produced by the Leuconostoc citreum HE29 Isolated from Traditional Fermented Pickle"

_molecules, 2023, doi:10.3390/molecules28207149_

Round 1

Reviewer 1 Report

Lactic Acid Bacteria (LAB) strains Leuconostoc citreum HE29 were isolated from traditional fermented pickles, Leuconostoc citreum HE29 can produce exopolysaccharide EPS HE29 , 7.04 ± 0.3 g L_1, Glucose was the only sugar monomer in the core unit of EPS HE29  detected  by HPLC analysis and glucan HE29 revealed 7.3 kDa of molecular weight. Structural characterization of glucan HE29 by 1H and 13C NMR spectroscopy analysis demonstrated that EPS HE29 was a dextran type EPS containing 5.3% levels of (1 → 3)-linked α-D-glucose units.  thermal properties detected by TGA and DSC analysis, dextran HE29 demonstrated a degradation temperature of around 280℃, showing its strong thermal features. A semi-crystalline nature was observed for dextran HE29 detected by XRD analysis.

I think  this manuscript can be published after the authors make some revise. My revision opinions are as follows:

1. In Figure 2. HPLC chromatogram of EPS HE29 , there are have two big peak, what is first peak (retention time about 7min)? is  it not a sugar monome? the author should give some detail illustration.

2. a molecular weight of 7.3 kDa was calculated by GPC analysis,  this molecular weight is very low, the author should supply GPC chromatogram.

Author Response

Lactic Acid Bacteria (LAB) strains Leuconostoc citreum HE29 were isolated from traditional fermented pickles,  Leuconostoc citreum HE29 can produce exopolysaccharide EPS HE29 , 7.04 ± 0.3 g L_1, Glucose was the only sugar monomer in the core unit of EPS HE29  detected  by HPLC analysis and glucan HE29 revealed 7.3 kDa of molecular weight. Structural characterization of glucan HE29 by 1H and 13C NMR spectroscopy analysis demonstrated that EPS HE29 was a dextran type EPS containing 5.3% levels of (1 → 3)-linked α-D-glucose units.  thermal properties detected by TGA and DSC analysis, dextran HE29 demonstrated a degradation temperature of around 280℃, showing its strong thermal features. A semi-crystalline nature was observed for dextran HE29 detected by XRD analysis.

I think this manuscript can be published after the authors make some revise. My revision opinions are as follows:

Dear Reviewer, Thank you very much for your comments that will enhance the quality of my manuscript. Please find my responses below.

1. In Figure 2. HPLC chromatogram of EPS HE29, there are have two big peak, what is first peak (retention time about 7min)? is it not a sugar monomer? the author should give some detail illustration .

Thank you very much for this comment. We hydrolyse the EPS through acid hydrolyses at high temperature followed by the neutralisation of the mixture with a basic solution results in the formation of a salt. The first peak is originating from this salt formed during hydrolysis step which I have now discussed in the manuscript (Line 112-113).

2. a molecular weight of 7.3 kDa was calculated by GPC analysis, this molecular weight is very low, the author should supply GPC chromatogram.

Thank you very much for this comment. I have added the GPC chromatogram as Figure 2B (Line 104).  

Reviewer 2 Report

The authors isolated Leuconostoc citreum HE29 from traditional fermented pickles, and determined the physicochemical and techno-functional properties of its exopolysaccharide (EPS). The study was well designed and the results suggested that this EPS HE29 exhibits the potential of being used as a biothickener agent in food formulations. Overall, the paper is well-written and the flow of content is acceptable. The manuscript can be accepted after the authors can address following issues after minor revisions:

Line 58-67: The authors don’t need to provide detailed information about methods in the section of Introduction, but should clearly indicate the aim and significance of this study.

FTIR should be revised to FT-IR. For example, line 152 and 171.

Line 170, “cm-1”.

The legend of Figure 5 should be revised.

Minor editing of English language required

Author Response

Comments and Suggestions for Authors

The authors isolated Leuconostoc citreum HE29 from traditional fermented pickles, and determined the physicochemical and techno-functional properties of its exopolysaccharide (EPS). The study was well designed and the results suggested that this EPS HE29 exhibits the potential of being used as a biothickener agent in food formulations. Overall, the paper is well-written and the flow of content is acceptable. The manuscript can be accepted after the authors can address following issues after minor revisions:

Dear Reviewer, Thank you very much for your comments that will enhance the quality of my manuscript. Please find my responses below.

Line 58-67: The authors don’t need to provide detailed information about methods in the section of Introduction, but should clearly indicate the aim and significance of this study.

Thank you very much for this comment. I have added the aim and significance of the study to the introduction section (Line 58-60).

FTIR should be revised to FT-IR. For example, line 152 and 171.

Done.

Line 170, “cm-1”.

Done.

The legend of Figure 5 should be revised.

Done.

Comments on the Quality of English Language

Minor editing of English language required

Done.

Reviewer 3 Report

I think that the article entitled Physicochemical characterization of dextran HE29 produced by the Leuconostoc citreum HE29 isolated from traditional fermented pickle is t adequate to be published in the journal.

I think that the research article is adequate to be published in the journal; however, it is necessary to adjust throughout the text.

1. Page 2, line 49 to 53. The text lacks bibliographical support, likewise, the technological relevance is briefly described, it should enhance this importance since in my opinion it would give greater relevance to the work. I suggest the following references:

· Castilla-Marroquín, J. D., Hernández-Martínez, R., de la Vequia, H. D., Ríos-Corripio, M. A., Hernández-Rosas, J., López, M. R., & Hernández-Rosas, F. (2020). Dextran synthesis by native sugarcane microorganisms. Revista Mexicana de Ingeniería Química,19(Sup. 1), 177-185.

· Hernández-Rosas, F., Castilla-Marroquín, J. D., Loeza-Corte, J. M., Lizardi-Jiménez, M. A., & Martínez, R. H. (2021). The importance of carbon and nitrogen sources on exopolysaccharide synthesis by lactic acid bacteria and their industrial importance. Revista Mexicana de Ingeniería Química, 20(3), Bio2429-Bio2429.

2. Figure 4. FTIR, It is necessary to identify the peaks and include the reference standard.

3. It is necessary to describe in more detail the system where the EPS was produced, as well as the production conditions; geometry, with or without agitation, cell adjustment, etc. It should be described in detail so that the system can be replicated.

4. At least 30 percent of the references are very old and should be updated as much as possible.

Author Response

Comments and Suggestions for Authors

I think that the article entitled Physicochemical characterization of dextran HE29 produced by the Leuconostoc citreum HE29 isolated from traditional fermented pickle is t adequate to be published in the journal.

I think that the research article is adequate to be published in the journal; however, it is necessary to adjust throughout the text.

Dear Reviewer, Thank you very much for your comments that will enhance the quality of my manuscript. Please find my responses below.

1. Page 2, line 49 to 53. The text lacks bibliographical support, likewise, the technological relevance is briefly described, it should enhance this importance since in my opinion it would give greater relevance to the work. I suggest the following references:

  • Castilla-Marroquín, J. D., Hernández-Martínez, R., de la Vequia, H. D., Ríos-Corripio, M. A., Hernández-Rosas, J., López, M. R., & Hernández-Rosas, F. (2020). Dextran synthesis by native sugarcane microorganisms. Revista Mexicana de Ingeniería Química,19(Sup. 1), 177-185.
  • Hernández-Rosas, F., Castilla-Marroquín, J. D., Loeza-Corte, J. M., Lizardi-Jiménez, M. A., & Martínez, R. H. (2021). The importance of carbon and nitrogen sources on exopolysaccharide synthesis by lactic acid bacteria and their industrial importance. Revista Mexicana de Ingeniería Química, 20(3), Bio2429-Bio2429.

Added.

2. Figure 4. FTIR, It is necessary to identify the peaks and include the reference standard.

Thank you very much for this comment. So far, FT-IR data from different glucan and dextran structures were reported in detail in the literature and I have compared the FT-IR peaks with the published literature data and I have used this data as reference to compare peak by peak.

3. It is necessary to describe in more detail the system where the EPS was produced, as well as the production conditions; geometry, with or without agitation, cell adjustment, etc. It should be described in detail so that the system can be replicated.

Thank you very much for this comment. I have added the culture conditions and the required information to the MM section.

4. At least 30 percent of the references are very old and should be updated as much as possible.

Thank you very much for this comment. I have given 5 papers from 2005, 2008, 2009, 2010 and 2006 and the others are from 2015 to today.  The former ones are important for the field and relevant to the manuscript. Thank you very much for your suggestion.

Reviewer 4 Report

Author described the properties of isolated daxtran which was produced by Leuconostoc citreum HE29 isolated from fermented pickles. There were some problems as followings.

1. Lines 40, 43, 44, 45….α-d-glucose units  were suggested to α-D-glucose units

2. It was suggested to add the data of 16S rRNA and RAPD-PCR of isolated HE29 as Leuconostoc citreum (Lines 75-78). It was suggested to analyze the difference of dextran HE 29 with dextran in literatures (such as 

(a) Int J Biol Macromol. 2018, 113:45-50. doi: 10.1016/j.ijbiomac.2018.02.119. Characterization of highly branched dextran produced by Leuconostoc citreum B-2 from pineapple fermented product;

(b) Carbohydr Polym. 2015, 133:365-72. doi: 10.1016/j.carbpol.2015.07.061. Isolation and characterization of dextran produced by Leuconostoc citreum NM105 from manchurian sauerkraut;

(c) Antioxidants 2023, 12(2), 275; https://doi.org/10.3390/antiox12020275. Characterization of Dextran Biosynthesized by Glucansucrase from Leuconostoc pseudomesenteroides and Their Potential Biotechnological Applications)

3. The entire methods for compositions and average molecular mass of dextran HE29 were not clear (please provide GPC data including standard markers and standard curve). It was suggested to describe more in details in method section, including hydrolysis conditions to get the monomer and  HPLC column conditions, and GPC standard curve and sample in GPC.

4. Figure 2 seemed contained several peaks. Please re-check and re-plot the figure.

5. Figure 3A contained higher amounts of alpha (1-6) linkage. Please calculate the percentages of alpha(1-6) in the dextran. It semmed a peak in the ppm4.7. What is the linkage type?

6. Lines 182-183...What is the " carboxyl group" in the xxopolysaccharide?     ........served originating from the moisture loss and this initial phase was previously suggested to be related with the high level of carboxyl group within the EPS structure

7. as line 188 (...260-320°C revealing the thermal degradation)...however, the sharp phase changes was not observed in the DSC (Figure 5B)?

Author Response

Dear Reviewer, Thank you very much for your comments that will enhance the quality of my manuscript. Please find my responses below.

Author described the properties of isolated daxtran which was produced by Leuconostoc citreum HE29 isolated from fermented pickles. There were some problems as followings.

1. Lines 40, 43, 44, 45….α-d-glucose units were suggested to α-D-glucose units

Done.

2. It was suggested to add the data of 16S rRNA and RAPD-PCR of isolated HE29 as Leuconostoc citreum (Lines 75-78).

Thank you very much for this comment. We have already added the 16S rRNA sequence data to the manuscript (Line 86); Please see the below;

The GenBank accession numbers of the 16S rRNA gene sequences of strains VB113, VK1, HE29 and VK36 were OQ801446-OQ801449

It was suggested to analyze the difference of dextran HE 29 with dextran in literatures (such as 

(a) Int J Biol Macromol. 2018, 113:45-50. doi: 10.1016/j.ijbiomac.2018.02.119. Characterization of highly branched dextran produced by Leuconostoc citreum B-2 from pineapple fermented product;

(b) Carbohydr Polym. 2015, 133:365-72. doi: 10.1016/j.carbpol.2015.07.061. Isolation and characterization of dextran produced by Leuconostoc citreum NM105 from manchurian sauerkraut;

(c) Antioxidants 2023, 12(2), 275; https://doi.org/10.3390/antiox12020275. Characterization of Dextran Biosynthesized by Glucansucrase from Leuconostoc pseudomesenteroides and Their Potential Biotechnological Applications)

Thank you very much for these comments. We have already discussed these papers in the manuscript as can be seen below:

‘For instance, Leuc. citreum strains NM105 from sauerkraut, SK24.002 from traditional pickled vegetables and B-2 from fermented pineapple were shown to produce a highly α-(1 → 2) branched dextran [19], a highly α-(1 → 3) branched dextran [16] and dextran containing α-(1 → 3)/α-(1 → 2) linked d-glucopyranose units [20]’

3. The entire methods for compositions and average molecular mass of dextran HE29 were not clear (please provide GPC data including standard markers and standard curve). It was suggested to describe more in details in method section, including hydrolysis conditions to get the monomer and  HPLC column conditions, and GPC standard curve and sample in GPC.

Thank you very much for this comment. I have added the GPC data as Figure 2B including the data as well the standards molecular weights. Additionally, I have added the HPLC conditions including the hydrolysis step to the MM section.

4. Figure 2 seemed contained several peaks. Please re-check and re-plot the figure.

Explained in the manuscript.

5. Figure 3A contained higher amounts of alpha (1-6) linkage. Please calculate the percentages of alpha(1-6) in the dextran. It semmed a peak in the ppm4.7. What is the linkage type?

Thank you very much for this comment. I have calculated 5.3% levels of (1 → 3)-linked α-D-glucose units and rest of the polysaccharide is formed by (1 → 6)-linked α-D-glucose units (94.7%). The peak at 4.7 is coming from the solvent as given in the MM section (D2O). I have given the linkage type in the manuscript. Thank you.

6. Lines 182-183...What is the " carboxyl group" in the xxopolysaccharide? ........served originating from the moisture loss and this initial phase was previously suggested to be related with the high level of carboxyl group within the EPS structure

Thank you very much for this comment. This was a suggestion but I have now discarded this suggestion from the manuscript as I have not proven the acidic structure of dextran HE29. Thank you.

7. as line 188 (...260-320°C revealing the thermal degradation)...however, the sharp phase changes was not observed in the DSC (Figure 5B)?

Thank you very much for this comment. This is due to the nature of the analysis in DSC and no sharp reduction can be seen but as can be seen in Figure 5B, a gradual degradation was observed at same temperatures. This data was also consistent with literature data.

Round 2

Reviewer 4 Report

1. Figure 2 (B), legend. The GPC marked three area 1, 2, and 3 in the figure. The major peak is located at area 2.  However, authors described that First peak originates from the EPS HE29 whereas the 2nd and 3rd peaks are the column impurity..

........GPC chromatogram of EPS HE29 showing the molecular weight of EPS HE29 as 7.3 kDa (B). First peak originates from the EPS HE29 whereas the 2nd and 3rd peaks are the column impurity.. 

2. Authors described the hydrolysis conditions for sugar compositions. It seeded that the high temperature and high concentration of sulfuric acid. The skeleton of Glc should be changed rather a single peak in the Figure 2A.

Author Response

Dear Reviewer Thank you very much for your comments that will enhance the quality of my manuscript. Please find my responses below.

1. Figure 2 (B), legend. The GPC marked three area 1, 2, and 3 in the figure. The major peak is located at area 2.  However, authors described that First peak originates from the EPS HE29 whereas the 2nd and 3rd peaks are the column impurity..

........GPC chromatogram of EPS HE29 showing the molecular weight of EPS HE29 as 7.3 kDa (B). First peak originates from the EPS HE29 whereas the 2nd and 3rd peaks are the column impurity.. 

 Corrected

2. Authors described the hydrolysis conditions for sugar compositions. It seeded that the high temperature and high concentration of sulfuric acid. The skeleton of Glc should be changed rather a single peak in the Figure 2A.

Thank you very much for this comment. We have written the methodology as suggested. Two peaks appeared as discussed in the manuscript one from the salt as impurity and the second one from the glucose monomer. Thank you.
